# Progress on the Application of Bortezomib and Bortezomib-Based Nanoformulations

**DOI:** 10.3390/biom12010051

**Published:** 2021-12-30

**Authors:** Jianhao Liu, Ruogang Zhao, Xiaowen Jiang, Zhaohuan Li, Bo Zhang

**Affiliations:** School of Pharmacy, Weifang Medical University, Weifang 261053, China; 17852062086@163.com (J.L.); 15264626411@163.com (R.Z.); sjvane0532@163.com (X.J.); lzh15098737363@163.com (Z.L.)

**Keywords:** bortezomib, proteasome inhibitor, cancer therapy, nanoformulations

## Abstract

Bortezomib (BTZ) is the first proteasome inhibitor approved by the Food and Drug Administration. It can bind to the amino acid residues of the 26S proteasome, thereby causing the death of tumor cells. BTZ plays an irreplaceable role in the treatment of mantle cell lymphoma and multiple myeloma. Moreover, its use in the treatment of other hematological cancers and solid tumors has been investigated in numerous clinical trials and preclinical studies. Nevertheless, the applications of BTZ are limited due to its insufficient specificity, poor permeability, and low bioavailability. Therefore, in recent years, different BTZ-based drug delivery systems have been evaluated. In this review, we firstly discussed the functions of proteasome inhibitors and their mechanisms of action. Secondly, the properties of BTZ, as well as recent advances in both clinical and preclinical research, were reviewed. Finally, progress in research regarding BTZ-based nanoformulations was summarized.

## 1. Introduction

Bortezomib (BTZ; trade name: Velcade^®^), an antineoplastic agent developed by the Millennium Corporation in the United States of America, is the first proteasome inhibitor approved by the Food and Drug Administration [1,2,3,4]. This drug can block the ubiquitin-proteasome system (UPS) pathway, thereby hindering the degradation of antitumor-related protein products and exerting an antitumor effect [5,6,7]. Currently, BTZ has been approved for the treatment of multiple myeloma (MM) and mantle cell lymphoma (MCL) [8]. BTZ also showed inhibitory effects on other hematological cancers and numerous solid tumors in preclinical studies and clinical trials [9,10,11]. Nevertheless, due to its low solubility, poor specificity, and significant toxicity, the application of BTZ in solid tumors is limited [12,13,14,15]. With the development of nanomaterials, a series of BTZ-based drug delivery systems have been developed to improve its therapeutic efficacy [16,17,18,19]. In this review, we summarized the functions of proteasome inhibitors and their mechanisms of action, the clinical and preclinical applications of BTZ, and the progress in research concerning BTZ-based nanoformulations.

## 2. The UPS and Proteasome Inhibitors

The UPS and lysosomal autophagy are the major metabolic pathways in eukaryotic cells, which can degrade a variety of protein products and damage senescent organelles [20,21,22,23]. The former plays an important role in regulating the biological processes of cells and maintaining homeostasis, it also participates in the hydrolysis of >80% proteins in cells [24,25]. It has been demonstrated that many proteins or cytokines involved in cell growth, proliferation, apoptosis, gene expression, and immune response are degraded by the UPS pathway [26,27]. 

The UPS pathway is the key pathway through which proteasome inhibitors exert their pharmacological effects. Proteasome inhibitors work by blocking the UPS, which is closely related to the degradation of caspase, the blockage of the nuclear factor-kappa B (NF-κB) signal pathway, the regulation of B-cell lymphoma-2 (Bcl-2) protein, the promotion of the expression of tumor suppressor genes, etc., [28,29,30,31,32,33]. 

### 2.1. Functions of the UPS

The degradation process of the UPS consists of two steps: substrate ubiquitin labeling and ubiquitinated protein degradation by proteasomes [34,35]. The lysine residues of substrate protein are labeled by forming isopeptide bonds with ubiquitin under the action of ubiquitin-activating enzyme (E1), ubiquitin-binding enzyme (E2), and ubiquitin-linker enzyme (E3). Finally, the residues are degraded by the 26S proteasome. The 26S proteasome consists of a 20S catalytic core and two 19S regulatory subunits that recognize ubiquitination substrates. The 20S catalytic core consists of four rings formed by two groups of α and β subunits [36]. Seven different α subunits form two outer rings, while seven different β subunits form two inner rings. The α subunit ensures the unfolded protein enters the catalytic core, while the β subunits have catalytic activity to hydrolyze the protein substrate [37]. β subunits can be further divided into different types; for example, β1 has caspase-like activity with the ability to dissociate acid residues, β2 has trypsin-like activity and can remove alkaline residues, and β5 has chymotrypsin-like activity and can lyse hydrophobic residues [38,39,40]. 

UPS abnormalities are closely related to the occurrence, development, and prognosis of malignant tumors. Incorrect degradation of target proteins by the UPS can lead to the aggregation of oncoproteins and abnormal degradation of tumor suppressor proteins. These effects will cause apoptotic blockage and accelerated proliferation of mutant cells, ultimately inducing the occurrence of tumors [41,42].

### 2.2. Proteasome Inhibitors

With the in-depth understanding of the UPS pathway, proteasome inhibitors have become a reliable option for the treatment of cancer. In the past few decades, a variety of proteasome inhibitors have been developed, including boric acid peptide, epoxy-based peptide, and non-peptide lactones [43]. These inhibitors effectively block the UPS pathway by covalently binding to the β subunit of the 20S core [44]. 

Boric acid peptide and epoxy-peptide are the most thoroughly studied proteasome inhibitors and have been successfully marketed. BTZ is a representative drug of boric acid proteasome inhibitors. In these agents, boric acid (as the pharmacophore group) can reversibly bind with threonine residues on the β subunit to form a tetrahedral complex, thereby effectively blocking the UPS [45,46]. However, epoxy-peptide drugs differ from boric acid peptide proteasome inhibitors. The epoxy ketone group of drugs can form a stable morpholine ring structure with hydroxyl and amino groups of proteasome Thrl-1. This permits irreversible binding to the proteasome, thus avoiding the off-target effects of boric acid proteasome inhibitors [47]. Currently, there are three commercially available proteasome inhibitors, namely BTZ, carfilzomib, and ixazomib. In addition, various other compounds (e.g., oprozomib, delanzomib, marizomib, etc.) are currently under investigation in clinical trials. Some representative proteasome inhibitors are summarized in Table 1.

### 2.3. Mechanisms of Proteasome Inhibitors

In addition to hematological cancers (e.g., myeloma), the UPS is extremely active in solid tumors (e.g., prostate and breast cancer) [54,55,56]. Accordingly, proteasome inhibitors inhibit the function of the UPS, thus inducing the accumulation of antitumor-related protein products (e.g., tumor suppressors, cyclins, and apoptosis suppressors). Various signaling pathways are involved in the antitumor functions of proteasome inhibitors (Figure 1), which mainly include: (i) blockage of the activation of the NF-κB signaling pathway (currently the most recognized mechanism of action) [57,58], (ii) induction of mitochondrial endogenous apoptosis (change in the permeability of the mitochondrial membrane leads to the release of apoptotic proteins from mitochondria) [59], (iii) promotion of the expression of tumor suppressor genes p53, p21, and p27 [60,61], (iv) activation of the endoplasmic reticulum stress (ERS)-mediated apoptosis pathway, and (v) activation of the c-Jun NH2 terminal kinase (JNK) pathway [62].

#### 2.3.1. Blockage of the NF-κB Signaling Pathway

NF-κB is an anti-apoptotic factor closely related to tumor growth, and the earliest verified antitumor pathway linked to proteasome inhibitors [63]. It is most commonly present as a dimer of p65-p50. I kappa B (IκB), an inhibitor protein of the p65-p50 heterodimer, can inactivate NF-κB [64,65]. Following the stimulation of cells by cytokines, antigens, viruses, etc., signal transduction is triggered to promote the phosphorylation and ubiquitination of IκB. Subsequently, ubiquitinated IκB is degraded by the proteasome, resulting in the release of NF-κB and translocation into the nucleus. NF-κB binds to promoter regions of genes encoding anti-apoptotic factors, thereby increasing the expression of anti-apoptotic proteins. Proteasome inhibitors, such as BTZ, could block the degradation of IκB by inhibiting the activity of the UPS. This process leads to inhibition of the activation of anti-apoptotic factor NF-κB [66]. The loss of NF-κB activity inhibits the survival and proliferation of tumor cells, thereby exerting antitumor effects.

#### 2.3.2. Regulation of Bcl-2 Protein and the Signal Transduction Pathway 

The mitochondrial endogenous pathway, which is mediated by the Bcl-2 protein family, plays a key role in cell apoptosis [67]. Bcl-2 protein, which is located in the membrane of mitochondria, nucleus, and endoplasm, regulates cell apoptosis. Members of the Bcl-2 protein family can be divided into three categories according to their structures: (i) anti-apoptotic proteins, including Bcl-2, Bcl-XL, and Bcl-w, (ii) pro-apoptotic proteins, mainly including Bcl-2 proteins, Bcl-2-associated X protein (Bax), and Bcl-2 antagonist killer 1 (Bak), and (iii) special members of pro-apoptotic proteins, including Bcl-2-associated agonist of cell death (Bad), Bcl-2-interacting killer (Bik), Bcl-2 homology domain 3 (BH3)-interacting domain death agonist (Bid), Bcl-2-interacting mediator of cell death (Bim), Bcl-2-modifying factor (Bmf), etc. All of these proteins have homologous sequences in the BH3 domain. Under the stress induced by proteasome inhibitors, the homologous BH3 domain acts as a sensing element to directly inhibit anti-apoptotic proteins or activate pro-apoptotic proteins (e.g., Bax). This increases the permeability of the mitochondrial membrane, leading to the release of cytochrome C and the activation of caspase-3. Finally, the caspase cascade leads to apoptosis. Letai et al. found that proteasome inhibitor BTZ could inhibit tumor growth. This effect was accompanied by decreased expression of Bcl-2 and Bcl-XL, enhanced expression of Bax, and activation of caspase-3. The investigators confirmed that BTZ could inhibit tumor growth through the mitochondrial apoptosis pathway [6]. 

#### 2.3.3. Induction of ERS

ERS occurs when protein accumulation exceeds the rate of protein folding or degradation. Tumor cells are strongly dependent on ERS and can maintain their stability through self-compensation of ERS [68]. Proteasome inhibitors can inhibit the degradation of proteins by blocking proteasome activity, resulting in the accumulation of a large number of proteins in cells [69]. This intracellular protein overload breaks the self-compensatory balance of tumor cells, increasing ERS. Consequently, ERS promotes the activation of apoptosis-related proteins, such as caspase-12 and calpain, and leads to cell apoptosis [6,70].

#### 2.3.4. Promotion of the Expression of Tumor Suppressor Genes

The p53 gene is a tumor suppressor gene in humans, and its transcription product (i.e., p53 protein) exhibits a strong tumor suppressor effect. However, studies have found that p53 protein has a very short half-life and is rapidly degraded by the UPS. Interestingly, it has been shown that proteasome inhibitors directly inhibit tumor growth by blocking the UPS to reduce the degradation of p53 protein [71]. In addition to p53, proteasome inhibitors can also regulate the expression of cyclins, such as p27 and p21, thus inhibiting the cell cycle and inducing apoptosis [72,73].

#### 2.3.5. Activation of the JNK 

Following an increase in the levels of misfolded proteins, the JNK protein promotes cell death. The JNK pathway promotes apoptosis mainly through TNF-α and caspase-8. Inhibition of JNK activation prevents the release of cytochrome C and the activation of caspase-8, thereby inhibiting apoptosis. Proteasome inhibitors enable JNK accumulation by inhibiting JNK phosphorylation, while activating TNF-α and caspase-8 to promote tumor apoptosis [74,75].

## 3. Properties and Applications of BTZ

### 3.1. Physical and Chemical Properties

BTZ is a hydrophobic drug which is easily soluble in organic solvents; the poor solubility of BTZ in aqueous solution limits its application in clinical practice. BTZ has an aminoalkyl dipeptide boronic acid, which contains pyrazinyl, L-phenylalanine, and L-boroleucine [76,77]. 

The boronic acid is an important functional group of BTZ and the key to its therapeutic effects [4,78,79]. The boron atom is sp2 hybridized, has a defect center, and can interact with the electron-rich system [8,80]. Consequently, BTZ can covalently bind to the 20s chymotrypsin-like active site in the 26s proteasome (the hydroxyl group of Thr1 at the N-terminus) to form a four-sided adductor [81,82]. 

In a neutral or alkaline solution, the dissociated BTZ can undergo esterification with 1-2 or 1-3 adjacent diol groups, resulting in the formation of pH-sensitive boronic acid ester bonds [82]. In contrast, BTZ exhibits a stable planar triangular structure under acidic conditions, which is difficult to combine with other substances. In recent years, researchers have discovered that acidity is a typical characteristic of the tumor microenvironment [83]. Therefore, pH-sensitive and reversible borate ester bonds can be selected to achieve the localized release of BTZ and improve the bioavailability. In addition, it has been reported that borate ester bonds are highly sensitive to glucose, adenosine triphosphate, H_2_O_2_, and other substances in the biological body. This evidence provided new directions for the efficient release of BTZ [84,85].

### 3.2. Clinical Applications of BTZ

BTZ is approved for the first-line treatment of refractory MM and MCL [86]. The combination of BTZ with other antitumor drugs in the treatment of MM and MCL has been clinically implemented. Representative clinical therapeutic regimens include: (i) RVD (BTZ, lenalidomide plus dexamethasone), (ii) VD (BTZ plus dexamethasone), (iii) VCD (plus cyclophosphamide on the basis of VD), (iv) VR-CAP (BTZ, rituximab, cyclophosphamide, doxorubicin, and prednisone), and (v) RVD+D (plus daratumumab on the RVD regimen). Clinical data verified that BTZ-based combination regimens exerted beneficial therapeutic effects and improved the quality of life of patients with MCL and MM. 

Chang et al. used a therapeutic regimen termed VCR-CAVD (i.e., a combination of BTZ, rituximab, cyclophosphamide, vincristine, doxorubicin, and dexamethasone) for the treatment of MCL. This was followed by 2 years of consolidation and maintenance therapy. The follow-up survey demonstrated that treatment with the VCR-CAVD regimen was effective and well tolerated in treatment-naive patients with MCL. The rates of complete response (CR), overall response rate (ORR), and 3-year progression-free survival (PFS) were 68%, 95%, and 72%, respectively [87,88]. 

Another clinical study examined the efficacy of BTZ or high-dose dexamethasone in patients with recurrent MM after receiving one to three lines of treatment. The 669 patients involved in this study were randomly assigned to receive BTZ alone or a combination of dexamethasone and BTZ. Surprisingly, monotherapy with BTZ was associated with better prognosis compared with the combination therapy (1-year survival: 80% vs. 66%; partial response: 38% vs. 18%; and CR: 6% vs. 1%, respectively [89]).

### 3.3. Clinical Trials of BTZ

The potential effectiveness of BTZ on other types of cancer has been investigated extensively in clinical trials and preclinical studies. Among those, hematological cancers are the most widely studied, mainly including Waldenstrom’s macroglobulinemia (WM), diffuse large B-cell lymphoma (DLBCL), peripheral T-cell lymphoma (PTCL), and acute myelogenous leukemia (AML). Additionally, the role of BTZ in the treatment of solid tumors (e.g., colon, non-small cell lung, and breast cancer) has been evaluated. Some representative clinical research is summarized in Table 2.

#### 3.3.1. Hematological Cancers

Hematological cancers, also termed blood tumors, mainly originate in the bone marrow or lymphatic tissue, they include WM, DLBCL, PTCL, AML etc. Blood tumors mainly exist in blood circulation and are characterized by rapid growth and poor response to surgical treatment. The effectiveness of BTZ in the treatment of MCL and MM offers promise for potential use against other hematologic tumors.

WM is a malignant hyperproliferative disease of inert B lymphocytes, characterized by massive secretion of monoclonal immunoglobulin M (IgM). In a phase II clinical study, 59 patients with WM were treated with a combination of BTZ, dexamethasone, and rituximab (BDR). The BDR regimen showed excellent therapeutic activity with good tolerance and no obvious myelotoxicity. The rate of CR, PR, and 3-year survival rate were 3%, 58%, and 81%, respectively. Progression-free survival (PFS) was 42 months. In addition, it can induce a persistent response in previously untreated patients with WM. 

Another clinical trial examined the use of BTZ in combination with everolimus and rituximab (BER) for the treatment of relapsed and refractory macroglobulinemia. Ghobrial et al. investigated the safety and effectiveness of this regimen. The results showed that this treatment was well tolerated, and the median PFS was prolonged. The rate of CR, PR, and PFS were 5.6%, 47.2%, and 21 months, respectively. This evidence suggests that the BER regimen is a new model for the clinical treatment of WM [90]. 

DLBCL is a type of non-Hodgkin’s lymphoma. A phase II trial of the CHOP (cyclophosphamide, doxorubicin, vincristine, and prednisone) regimen in combination with rituximab and incremental doses of BTZ was carried out in patients with DLBCL. The ORR, 2-year PFS, and 2-year overall survival rates were 88%, 64%, and 70%, respectively [91]. 

PTCL is a heterogeneous aggressive lymphoma derived from mature T cells and natural killer cells. In a phase I clinical trial, BTZ combined with CHOP was used to treat 46 patients with PTCL. The results showed that the ORR, CR rate, and overall 3-year survival rate associated with the combined chemotherapy regimen were 76%, 65%, and 47%, respectively, indicating significant therapeutic effects. Moreover, Fader et al. reported that standard doses of idarubicin, cytarabine, and BTZ were effective in 83% of patients with AML, and the CR rate was 58% [92].

#### 3.3.2. Solid Tumors

In recent years, studies on BTZ have provided a new approach to the treatment of solid tumors. A phase II study investigated the efficacy of a combination of BTZ and pegylated liposomal doxorubicin in patients with metastatic breast cancer. Patients received BTZ through intravenous administration on days one, four, eight, and eleven, and doxorubicin on days one and eight of the 21-day treatment cycle (partial response: 8%; median overall survival: 4.3 months). Due to the limited number of trials, the efficacy of BTZ combined with pegylated liposomal doxorubicin in the treatment of metastatic breast cancer remains to be explored [93].

A phase II trial was performed to investigate the effects of BTZ in combination with docetaxel in 80 patients with advanced non-small cell lung cancer. The 1-year survival rate was 33%, and the median survival was 7.8 months. This combination regimen was more effective than monotherapy with BTZ [94]. 

### 3.4. Pharmacokinetics of BTZ

The pharmacokinetics of BTZ are characterized by a two-compartment model. BTZ has a rapid initial distribution stage with an elimination half-life of >40 h [95,96]. Following intravenous administration, BTZ can be rapidly distributed throughout the whole body, except the brain and adipose tissue; of note, relatively high concentrations of the drug are detected in the liver and gastrointestinal tract [97]. The pharmacokinetic properties of BTZ were confirmed by Reece and Moreau et al. in the treatment of relapsed MM [98,99]. After one or more repeated administrations, the distribution volume of BTZ was 498–1884 L/m^2^, the plasma protein binding rate was 83%, and the total body clearance rate was 1095–1866 mL/min. BTZ is mainly metabolized by oxidative deborization of the cytochrome P450 enzyme and is completely cleared 72 h after injection [96,100]. 

The safety, tolerability, and pharmacokinetic characteristics of BTZ in patients with different degrees of renal and liver impairment were evaluated. At the dosage of 1.3 mg/m^2^, BTZ was well tolerated without significant changes observed on the maximum plasma concentration and half-life [101,102]. Moreau et al. investigated the pharmacokinetics of BTZ after subcutaneous and intravenous administration. The data showed that the systemic exposure and therapeutic effects of the two approaches were similar. Nevertheless, subcutaneous injection is favored due to the lower incidence of blood toxicity and peripheral neuropathy [58].

### 3.5. Adverse Reactions and Drug Resistance Induced by BTZ

BTZ exerts significant therapeutic effects on both hematological cancers and solid tumors. Nonetheless, the toxicity and side effects of BTZ cannot be ignored. Representative adverse drug reactions associated with BTZ mainly include: (i) fatigue (the most common adverse reaction), (ii) digestive tract reaction (nausea, diarrhea, constipation, vomiting, etc.) [103,104,105], (iii) peripheral neuropathy (e.g., numbness, pain, and paresthesia) [106,107], (iv) thrombocytopenia, neutropenia, and anemia [108], and (v) hypotension (in some patients during the treatment period) [109].

Numerous studies have confirmed that BTZ can cause drug resistance during administration. Franke et al. found that mutations in the β subunit weakened the binding of BTZ to the β5 subunit, leading to the development of drug resistance in patients [110]. This may be attributed to mutations caused by BTZ in key sites of proteasome subunit beta 5 (PSMB5). PSMB5 encodes a protein termed the β5 subunit. Changes in PSMB5 lead to an increase in the β5 subunit, thereby counteracting the inhibitory effect of β5 binding to BTZ [111]. In addition, the researchers found that the expression of neural precursor cell expressed developmentally downregulated 4-1 (NEDD4-1) E3s (member of the NEED4 protein family) and the enhancement of autophagy may be the cause of BTZ resistance [112,113].

## 4. Research Progress Regarding BTZ-Based Nanoformulations

Currently, the disadvantages of BTZ (e.g., high hydrophobicity, poor stability, low bioavailability, high toxicity to normal tissues, and drug resistance) limit its clinical applications and efficacy. In addition, BTZ binds nonspecifically to proteins in serum and is rapidly cleared in the liver, impairing its accumulation and penetration into solid tissues. This limits the use of BTZ for the treatment of solid tumors. 

With the rapid development of nanotechnology, drug delivery systems (e.g., liposomes, polymer micelles, inorganic nanoparticles, and biomimetic nanoparticles) have become a research hotspot (Figure 2). The nano-delivery platform can increase the accumulation of drugs at the targeted site and realize controlled drug release, thereby improving the therapeutic effects and reducing the occurrence of side effects. Thus far, numerous BTZ-based nanoformulations have been developed and investigated to further improve druggability and enhance the therapeutic effects (Table 3). In this section, research progress regarding BTZ-based nanoformulations is summarized.

### 4.1. Liposomes

Liposomes are closed vesicles with a bilayer structure, which are composed of phospholipid and cholesterol [132]. Owing to the similar biochemical structure to that of the cell membrane, liposomes have a favorable safety profile and biocompatibility [133]. Thus far, liposomes have been widely utilized for the delivery of drugs, and numerous products have become commercially available. Hydrophilic and hydrophobic drugs can be wrapped into the hydrophilic core and phospholipid bilayer, respectively [134].

Vijay et al. prepared D1X-encapsulated BTZ (D1XB) liposomes, in which BTZ and dexamethasone were loaded. Dexamethasone, a cholesterol analogue, can be modified on the surface of the liposomes to strengthen and stabilize the lipid bilayer. In addition, dexamethasone is a synthetic ligand of glucocorticoid receptors that can specifically bind to tumor cells with high expression of these receptors [114]. Studies have shown that the loading in liposomes improves the accumulation of BTZ into tumor sites. D1XB has demonstrated strong cytotoxicity against WM cells.

Zuccari et al. prepared asparagine-glycine-arginine (NGR) peptide active-targeting liposomes, termed NGR-SL (amino-lactose-BTZ [LM-BTZ]) [26]. BTZ and LM were firstly complexed via a borate ester bond, and subsequently encapsulated into the liposomes. By forming pH-sensitive borate ester bonds, LM can bind the poorly water-soluble BTZ to the hydrophilic core and prevent leakage of free BTZ from the liposomes. This process improves the encapsulation efficiency and achieves specific release of the drug. In a mouse model of neuroblastoma, NGR-SL (LM-BTZ) exhibited a higher therapeutic index and lower toxicity than the free drug.

### 4.2. Polymeric Micelles

Polymer micelles are a type of nanoparticle with a core-shell structure formed by the self-assembly of amphiphilic block copolymers in aqueous solution. The hydrophobic core can accommodate the insoluble substances to improve solubility. Besides the physical encapsulation manner, the drug can also be chemically conjugated to the amphiphilic block copolymers to realize the drug loading and delivery [135]. 

Liu et al. designed a dual pH-sensitive drug delivery system termed BTZ-polymer conjugates (BTZ-PC). The copolymers were synthesized by catechol-functionalized PC conjugation with polyethylene glycol (PEG) via acetal bonds, followed by formation of the micelles after self-assembly [116]. BTZ can be encapsulated into the hydrophobic core of the micelles by conjugation with catechol via a borate ester bond. Both acetal and borate ester bonds are acid-unstable and can be broken in the acidic tumor microenvironment to achieve pH-sensitive drug release. In a xenograft mouse model of human breast cancer, BTZ-PC micelles exerted enhanced antitumor effects and reduced toxicity compared with the free drug.

Liu et al. modified dopamine methylacrylamide to the end of PEG-block-poly(D,L-lactic acid) (PEG-b-PCL) using the “graft” method, leading to the formation of the PEG-b-PCL-b-PDMA co-polymer. Afterwards, BTZ was grafted on the polymer chain by conjugation with the catechol group in dopamine to form a pH-sensitive borate ester bond. Subsequently, BTZ-loaded copolymer micelles were formed under self-assembly [117]. The borate ester bond is a dynamic covalent bond, which improves the therapeutic effects of the drug. In a mouse model of hepatocellular carcinoma, the micelles were associated with stronger antitumor effects and a lower incidence of adverse reactions compared with the free drug.

Gu et al. synthesized a novel single copolymer HA-P(TMC-co-DTC) and prepared the biodegradable micelles (HA-curcumin) by self-assembly. In this process, the shell of HA is crosslinked to the core by disulfide bonds to increase the stability of the micelles [118]. BTZ-loaded micelles (HA-core-disulfide-crosslinked biodegradable micelles-BTZ-pinanediol [HA-CCMS-BP]) were prepared by encapsulating alcohol-esterified BTZ in the carriers. HA can actively target solid tumors overexpressing CD44, while disulfide bonds can selectively release the drug in tumor sites with high glutathione expression. The results showed that the cytotoxicity of HA-CCMS-BP was lower than that of the free drug. In a mouse model of MM, HA-CCMS-BP targeted therapy for MM was better tolerated than the free drug. 

### 4.3. Dendrimers

Dendrimers refer to dendritic macromolecules that start from core molecules and repeatedly grow outwards to form highly branched structures [136] The cargoes encapsulated in dendrimers can stably exist due to the spatial effects. Furthermore, the groups on the surface of dendrimers can control the solubility of hybrid nanocomposites and be linked with other molecules for further modification. Dendritic materials, such as polyamidoamine (PAMAM) and branched polyetherimide, have been widely used as carriers for drug delivery [137].

Wang et al. used PEG as a linker to connect cyclic arginine-glycine-aspartic acid (cRGD) to fifth-generation PAMAM and performed modifications with catechol to obtain the nanocarriers DPA-G5-PEG-cRGD [120]. BTZ was subsequently conjugated with catechol by a pH-sensitive borate ester bond to obtain a novel DPA-G5-PEG-cRGD/BTZ formulation. cRGD can target integrin αvβ3, which is highly expressed during tumor angiogenesis, thus achieving effective drug internalization. In healthy mice, the blood toxicity associated with this formulation was lower than that observed with the free drug. Interestingly, cRGD-PEG-PAMAM-Cat/BTZ also showed significant antitumor activity in a mouse model of MM.

Similarly, Wang et al. prepared a novel nanoformulation (G5-KAC-Cat-BTZ), in which dopamine (Cat) functionalized with a neutral shell (acetylated lysine, KAC) was grafted onto G5 PAMAM. Next, BTZ was loaded into the dendrimers through reaction with catechol [121]. In the acidic tumor microenvironment, the borate ester bond formed by BTZ and Cat realized effective drug release.

### 4.4. Nanogel

Nanogel is a type of water-soluble, three-dimensional, gelatinous nanoformulation formed after the crosslinking of the polymers. The network structure of the nanogel provides convenient space for accommodating a variety of compounds, such as molecular drugs, luciferin, proteins, peptides, and nucleic acids. Advantages of the nanogel include adjustable particle size, large surface area, high water content, and degradability [138,139]. In recent years, the construction of a controlled drug release system using the nanogel has become a research hotspot.

Liu et al. developed a novel method for the encapsulation of BTZ, in which BTZ was loaded with dopamine-grafted HA nanogel via a borate ester bond [122]. Owing to the location of dopamine on the skeleton structure of the nanogel (NSS-BTZ), more BTZ loading was achieved (≤8.58%). In this nanoformulation, HA can actively target CD44, while the borate ester bond is pH-sensitive. These advantages enable accurate and efficient drug release in the acidic tumor microenvironment with high CD44 expression. In an animal model, free BTZ was linked to severe systemic toxicity. In contrast, at the same dosage, BTZ loaded into the nanogel exhibited better antitumor efficacy and was associated with a significantly lower rate of side effects.

In a study performed by Amin et al., the highly efficient photothermal agent pNIPAAm-co-pAAm was synthesized. Afterwards, dopamine nanoparticles (DP) as the carriers of BTZ were incorporated into pNIPAAm-co-pAAm to prepare the temperature-responsive nanogel, termed NIPAM-AM-DP [123]. The nanogel directly kills the tumor cells through photothermal therapy, as well as localizes and releases BTZ for combination therapy.

### 4.5. Inorganic Nanoparticles

The term inorganic nanoparticles refers to a category of nanocarriers based on inorganic materials. Their advantages include good size control, low toxicity, and a large specific surface area [140,141]. In addition, inorganic nanoparticles have general properties, such as broad availability, easy functionality, good biocompatibility, and controlled release. Many inorganic materials have been widely used in the development of nanodrugs for the diagnosis and treatment of tumors, such as calcium phosphate, gold nanoparticles, carbon materials, mesoporous silica, etc., [142].

Sabine et al. prepared mesoporous silica nanoparticles (MSNs) for the encapsulation of BTZ [126]. The heptapeptide (HP) was combined with anti-biotin protein to block the pores of MSNs, thus yielding a new formulation termed MSNsHP. HP can be specifically recognized and cleaved by matrix metallopeptidase 9 (MMP9), which is highly expressed in tumor tissue, thereby opening the MSN channel to release the drug. The results showed that MSNsHP could remain stable under normal conditions, while it responsively released BTZ in the acidic tumor microenvironment. As the nanocarrier for BTZ, MSNsHP reduced the systemic toxicity of BTZ and selectively released the drug.

Gozde et al. prepared chitosan-coated superparamagnetic iron oxide nanoparticles (CS-MNP) for the encapsulation of BTZ. The nanoparticles were formed by ionic crosslinking in the presence of chitosan and tripolyphosphate molecules [127]. It was found that CS-MNP improved the therapeutic efficacy of BTZ. Because the protonated amino group of chitosan produces a swelling of osmotic pressure at a low pH, CS-MNP has the advantage of pH-responsive release.

### 4.6. Biomimetic Nanomaterials

Biomimetic nanomaterials are “living” materials with similar functions to those of living organisms. Unlike allogenic compounds (e.g., chemically synthesized polymers or inorganic nanoparticles), biomimetic materials are associated with low toxicity and not recognized by the immune system. This may be beneficial for improving the pharmacokinetics and biodistribution of the drugs [143,144]. Biomimetic nanoparticles are generally developed by coating or mixing the biomimetic nanomaterials with organic or inorganic nanoparticles [145]. At present, biomimetic nanomaterials (e.g., protein polypeptides, DNA, RNA, exosomes, phages, various cells, and cell membranes) have been extensively studied. Among those, albumin, exosomes, and cell membranes are the most widely investigated materials. Specifically, it has been shown that a variety of cell membranes (erythrocyte, platelet, tumor, macrophage, neutrophil, etc.) are useful for drug encapsulation.

Yu et al. synthesized a copper sulfide/carbon dot nanocomposite (CuSCD) with a hollow structure. CuSCD could be utilized as a multifunctional nanocarrier for both the photothermal conduction and encapsulation of BTZ [130]. Furthermore, the macrophage membrane hybridized with the T7 peptide was coated on the surface of BTZ-loaded CuSCD to form a novel biomimetic drug delivery system termed CuSCDB@MMT7. The T7 peptide binds to the ferritin receptor, which is overexpressed on tumor cells, and mediates the endocytosis of the nanoparticles. In addition, the macrophage membrane avoids immune recognition, thus ensuring that the therapeutic drugs safely reach the target. CuSCDB@MMT7 exerts both phototherapeutic and chemotherapeutic effects and has exhibited greater antitumor efficacy in the breast cancer mouse model. 

Min et al. obtained phage P22 virus capsids by high-temperature treatment [131]. Catechol ligands were chemically attached to the interior surface of the P22 viral capsid for subsequent encapsulation of BTZ. A novel nanoplatform was obtained by connecting hepatocellular carcinoma-targeting peptide SP94 (SFSIIHTPILPL) as a ligating agent to the outer surface of the P22 virus capsid nanocomposite. This biomimetic system is characterized by high efficiency for the loading of BTZ. Owing to the presence of SP94, the biomimetic nanoparticles can achieve a localized release of BTZ.

## 5. Outlook

BTZ is the first proteasome inhibitor approved by the Food and Drug Administration. It has demonstrated unique advantages in the treatment of MCL and MM, and numerous BTZ-based combination regimens have been widely utilized in clinical practice. Moreover, according to preclinical and clinical data, BTZ showed high antitumor activity against different hematological cancers and solid tumors. With the development of nanomaterials and their application in drug delivery, BTZ-based nanoformulations have been developed and widely investigated to improve the antitumor efficacy and reduce the occurrence of side effects. Although great advances have been achieved, the current understanding of BTZ remains insufficient. Further research focusing on the crosstalk between different action pathways, the in vivo metabolic process, and the immune effects on solid tumors is warranted. In addition, the unfavorable safety profile of nanomaterials, unexpected accumulation in normal tissues, poor stability of nanoparticles, etc., pose barriers to the commercialization of BTZ-based nanoformulations. Therefore, further understanding of the mechanism of BTZ in vivo is necessary. Currently, the construction of novel nanomedicines with the desired safety and specific targeting properties remains a challenge. It is expected that this field of research will attract considerable attention in the future. 

## Figures and Tables

**Figure 1 biomolecules-12-00051-f001:**
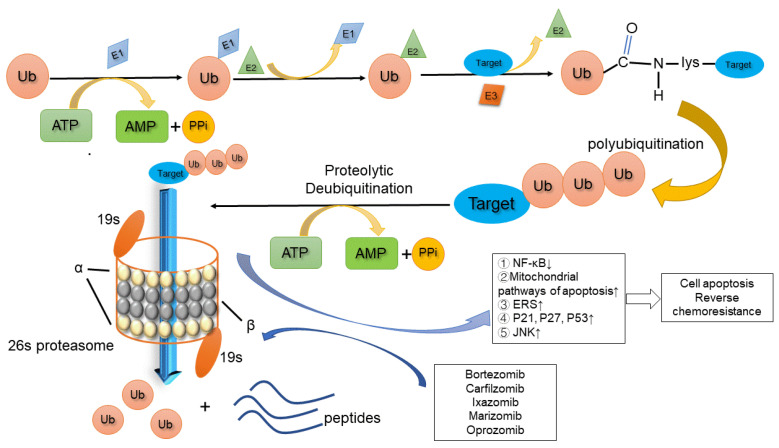
The target protein is ubiquitinated and transported to the 26S proteasome for degradation. Proteasome inhibitors bind to the β subunit of the catalytic core, thereby inhibiting the degradation of antitumor-related protein products. E1: ubiquitin-activating enzyme; E2: ubiquitin-binding enzyme; E3: ubiquitin-linker enzyme; Ub: ubiquitin.

**Figure 2 biomolecules-12-00051-f002:**
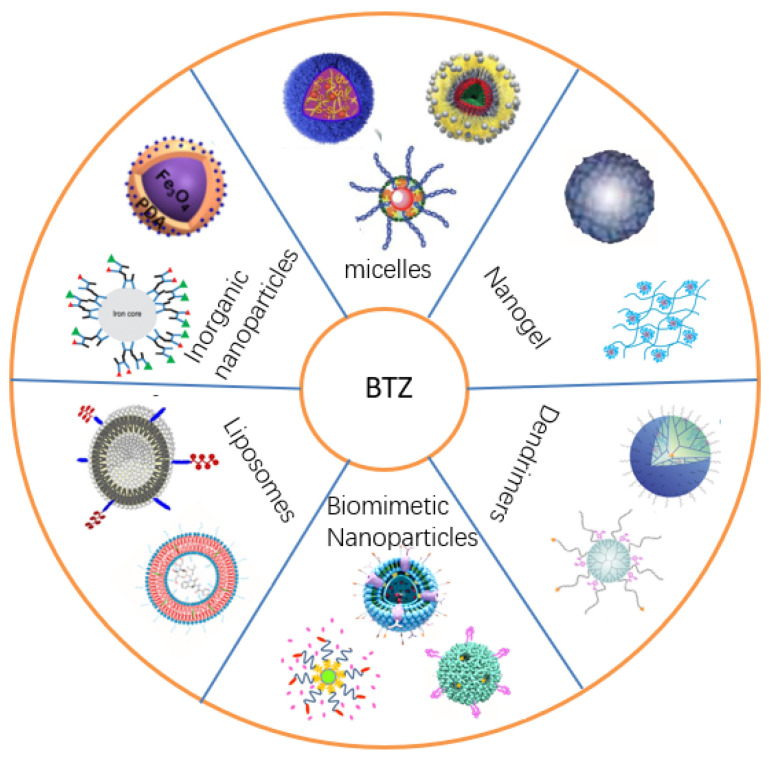
The nanocarriers for the delivery of BTZ.

**Table 1 biomolecules-12-00051-t001:** Proteasome inhibitors that are approved or currently investigated in clinical trials.

Proteasome Inhibitor	Company	Phase	Structure Type	Targets	Mechanism	Administration Route	Covalent Bond
Bortezomib	Millennium	Approved by the FDA in 2003	Dipeptide boronic acid	β1, β5	Covalently bonds with the N-terminal threonine of the β subunit	Intravenous	Reversible
Carfilzomib	Kyprolis	Approved by the FDA in 2012	Epoxy-peptide	β5	Combines with threonine at the active site of the β5 subunit to form a stable morpholine ring [48]	Intravenous	Irreversible
Ixazomib	Takeda	Approved by the FDA in 2015	Dipeptide leucine	β5	Forms a covalent bond with the N-terminal threonine of the trypsin-like active site [49,50]	Intravenous /oral	Reversible
Oprozomib	Amgen	Phase I/II	Tripeptide epoxy ketone	β5	Combines with threonine at the active site of the β5 subunit to form a stable morpholine ring [51]	Oral	Irreversible
Delanzomib	Teva	Phase I/II	Threonine Boric Acid	β1, β5	Forms a covalent bond with the N-terminal threonine of the trypsin-like active site [52]	Intravenous	Reversible
Marizomib	Celgene	Phase III	β-lactone-γ-lactam	β1, β2, β5	Thr1O γ on the subunit is covalently bound to the carbonyl group derived from the β-lactone ring of the inhibitor [53]	Intravenous	Irreversible

**Table 2 biomolecules-12-00051-t002:** The application of BTZ in clinics and clinical trials.

Disease	Regimen	Status	Outcomes
MCL	VCR-CAVD	Approved	CR 68%, ORR 95%, three-year PFS 72%
MM	BTZ	Approved	one-year survival was 80%, CR 6%, PR 38%
WM	BDR	Phase II	CR 3%, PR 58%, PFS 42 months, three-year survival rate 81%
WM	BER	Phase I/II	CR 5.6%, PR 47.2%, PFS 21 months
DLBCL	R-CHOP+BTZ	Phase II	ORR 88%, CR/Cru 75%, two-year overall survival rate 70%, two-year progression-free survival rate 64%
PTCL	CHOP+BTZ	Phase II	ORR 76%, CR 65%, three-year overall survival rate 47%, PFS 35%
AML	BTZ+idarubcin+ cytarabine	Phase II	The overall effective rate 83% CR 58%
Breast cancer	BTZ+ pegylated liposomal doxorubicin	Phase II	PR 8%, median overall survival 4.3 months
Lung cancer	BTZ+ docetaxel	Phase II	One-year survival was 33%, Disease control rates were 54%

**Abbreviations**: WM: Waldenstrom’s Macroglobulinemia; DLBCL: Diffuse Large B-cell Lymphoma; PTCL: Peripheral T-cell Lymphoma; AML: acute myelogenous leukemia; BDR: BTZ, dexamethasone and rituximab; BER: BTZ, everolimus and rituximab; R-CHOP: rituximab, cyclophosphamide, doxorubicin, vincristine, and prednisone; CHOP: cyclophosphamide, doxorubicin, vincristine, and prednisone; CR: complete response rate; PFS: progression-free survival; ORR: overall response rate; PR: partial response.

**Table 3 biomolecules-12-00051-t003:** Representative BTZ-based nanoformulations in cancer therapy.

Drug Delivery Systems	Nanocarriers	Drug	Ligand/Target	Cancers	Ref.
Liposomes	DODEAC, DEX, cholesterol	BTZ, DEX	DEX/GR	WM	[114]
Liposomes	PSGL-1, DPPC, Cholesterol	BTZ, Y27632	PSGL-1/P-selectin	MM	[115]
Liposomes	HSPC, Cholesterol, DSPE-PE2000-MAL	BTZ	NGR/aminopeptidase N	Neuroblastoma	[26]
Polymeric micelles	catechol-functionalized PC, PEG	BTZ	/	Breast cancer	[116]
Polymeric micelles	PEG-b-PCL	BTZ	/	Hepatocellular carcinoma	[117]
Polymeric micelles	HA-P(TMC-co-DTC)	BTZ	HA/CD44	Multiple myeloma	[118]
Polymeric micelles	DSPE-PEG, polydopamine	BTZ, DOX	/	Breast cancer	[119]
Dendrimers	PEG-PAMAM-Cat	BTZ	cRGD/Integrin αvβ3	Breast cancer	[120]
Dendrimers	G5-PAMAM-KAC-Cat	BTZ	/	Breast cancer	[121]
Nanogels	HA, acylate anhydride, dopamine	BTZ	HA/CD44	Hepatocellular carcinoma	[122]
Nanogels	pNIPAAm-co-pAAm	BTZ	/	Colon cancer	[123]
Nanogels	PLL−P(LP-co-LC), mPEG-b-PLL/DMMA	BTZ, CA4P	/	Colon cancer	[124]
Nanogels	P(Gu)5-PEG-P(Gu)5, P(Bor)5-PEG-(Bor)5,	BTZ	/	Multiple myeloma	[125]
Inorganic nanoparticles	MSNs	BTZ	HP/MMP2/9	Lung cancer	[126]
Inorganic nanoparticles	chitosan, magnetic iron oxide	BTZ	/	Breast cancer	[127]
Inorganic nanoparticles	mSiO_2_-H2A	BTZ	/	Cervical cancer	[128]
Inorganic nanoparticles	p(HEMA-co-DMA), superparamagnetic iron oxide	BTZ	/	Squamous-cell carcinoma	[129]
Biomimetic nanoparticles	macrophage membrane	BTZ	T7/TFR	Breast cancer	[130]
Biomimetic nanoparticles	phage P22 virus capsids	BTZ	/	Hepatocellular carcinoma	[131]

## Data Availability

Not applicable.

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
