# Peer review of "Progress on the Application of Bortezomib and Bortezomib-Based Nanoformulations"

_biomolecules, 2021, doi:10.3390/biom12010051_

Round 1

Reviewer 1 Report

The review entitled “Progress on the application of bortezomib and bortezomib- 2 based nanoformulations” by Liu and co-authors provides an interesting discussion about the clinical history of Bortezomib and the upcoming formulations it is formulated with to improve its therapeutic application. The structure of this review is good, but it demands some further improvement before being considered definitively suitable for publication.

  • An extensive revision by an English mother tongue is required, some sentences are very difficult to understand. Furthermore, there are a number of typo errors or misspelled words.
  • The authors commonly refer to BTZ as an UPS inhibitor. It is more appropriate to present it as a proteasome inhibitor that has an overall activity on UPS dynamics.
  • About NF-kB and IkBα. I acknowledge that the scientific community often refers to this pathway in such a way, but IkBα is the inhibitor of the p65-p50 heterodimer It is not appropriate to consider it the inhibitor of NF-kB which, instead, is a pathway encompassing several transcription factors unrelated to IkBα inhibition. transcription factors
  • Not all clinical studies introduced are provided with the relevant pharmacological and clinical information about the trial and primary (secondary) outcomes. This can be improved. With respect to this, Table II is well done. Something similar for the previously cited BTZ clinical application would be appreciated for the sake of readership.

Author Response

The review entitled “Progress on the application of bortezomib and bortezomib-based nanoformulations” by Liu and co-authors provides an interesting discussion about the clinical history of Bortezomib and the upcoming formulations it is formulated with to improve its therapeutic application. The structure of this review is good, but it demands some further improvement before being considered definitively suitable for publication.

1. An extensive revision by an English mother tongue is required, some sentences are very difficult to understand. Furthermore, there are a number of typo errors or misspelled words.

Response: Thanks for your comment. The full text was sent to a professional language company for English editing, and the certificate was attached. All the changes on language modifications were highlighted in blue color in the revised manuscript.

2. The authors commonly refer to BTZ as an UPS inhibitor. It is more appropriate to present it as a proteasome inhibitor that has an overall activity on UPS dynamics.

Response: We have corrected of the relationship between UPS and BTZ. The description was updated as “BTZ works by blocking the UPS pathway by inhibiting proteasome activity”.

3. About NF-kB and IkBα. I acknowledge that the scientific community often refers to this pathway in such a way, but IkBα is the inhibitor of the p65-p50 heterodimer It is not appropriate to consider it the inhibitor of NF-kB which, instead, is a pathway encompassing several transcription factors unrelated to IkBα inhibition. transcription factors.

Response: We agree with your opinion. The relationship between NF-KB and IkB have been well described in many studies[1,2]. Herein, we refined the expressions of NF-κB and IκB: "NF-κB is most commonly present as a p65-p50 dimer, and IκB is an inhibitory protein of p65-p50 heterodimer, which can bind to NF-κB to inactivate it".

The description was updated in red in the revised manuscript.

References:

[1] Markovina, S.; Callander, N.S.; O'Connor, S.L.; Kim, J.; Werndli, J.E.; Raschko, M.; Leith, C.P.; Kahl, B.S.; Kim, K.; Miyamoto, S. Bortezomib-resistant nuclear factor-kappaB activity in multiple myeloma cells. Mol Cancer Res 2008, 6, 1356-1364, doi:10.1158/1541-7786.Mcr-08-0108.

[2] Sánchez-Serrano, I. Success in translational research: lessons from the development of bortezomib. Nat Rev Drug Discov 2006, 5, 107-114, doi:10.1038/nrd1959.

4. Not all clinical studies introduced are provided with the relevant pharmacological and clinical information about the trial and primary (secondary) outcomes. This can be improved. With respect to this, Table II is well done. Something similar for the previously cited BTZ clinical application would be appreciated for the sake of readership.

Response: Thanks for your positive feedback. The relevant primary and secondary outcomes in both clinical applications (Section 2.2) and clinical trials (Section 2.3) was supplemented in the revision. Besides, the examples in Section 2.2 were also summarized into Table 2.

Special thanks for your insightful comments and suggestions.

Reviewer 2 Report

In this manuscript, the authors reviewed the applications of bortezomib as well as bortezomib-based nanoformulations. The topic was valuable and the whole content was well written, hereby I suggest it be accepted after minor revision. The main comments were given as below:

  1. Many abbreviations were used in the manuscript, while some of their full names were missing.
  2. In Figure 1, some texts were illegible and high-resolution picture are needed. Besides, the abbreviations such as Ub, E1, E2 should be specified.
  3. In section 2.4, more details on the pharmacokinetics of BTZ could be described, such as the oral bioavailability, drug-protein binding rate.
  4. Similarly, in section 2.5, the reasons on drug resistance of BTZ should be supplemented.
  5. In Figure2, PDA-Fe3O4 was drawn as an example. Has it been applied as the carrier of BTZ? The quality of Figure2 also needs to be improved.
  6. Some errors exists in the manuscript, and the full text needs to be checked thoroughly. For example:

1)    In Table 2, the regimen “BOR” should be “BDR”.

2)  The serial number of section “2. 2. Properties and applications of BTZ” should be section “2”.

3)    The name of the “NF-κB” should be written in consistent but not “NF-KB”.

Author Response

In this manuscript, the authors reviewed the applications of bortezomib as well as bortezomib-based nanoformulations. The topic was valuable and the whole content was well written, hereby I suggest it be accepted after minor revision. The main comments were given as below:

1. Many abbreviations were used in the manuscript, while some of their full names were missing and need to be clarified as they first appeared.

Response: Thanks for your pointing that out. We have reviewed and notified all the acronyms in the revised manuscript.

2. In Figure 1, some texts were illegible and high-resolution picture are needed. Besides, the abbreviations such as Ub, E1, E2 should be specified.

Response: Thanks for your suggestion. We have re-drawn the picture with high resolution and added the explanations about the acronyms in Figure 1.

3. In section 2.4, more details on the pharmacokinetics of BTZ could be described, such as the oral bioavailability, drug-protein binding rate.

Response: The detailed description of the pharmacokinetics of BTZ was supplemented in the revision.

Reece and Moreau et al confirmed the pharmacokinetic properties of BTZ in the treatment of relapsed multiple myeloma [1,2]. After one or repeated administration, the distribution volume of BTZ was 498-1884 L/m2, the plasma protein binding rate was 83%, and the total body clearance rate was 1095-1866 mL/min.

Concerning the oral bioavailability of BTZ, it is regrettable that no pharmacokinetic parameter related to oral absorption has been disclosed. It was suspected that oral absorption of BTZ was limited due to low aqueous solubility of the drug and efflux by P-glycoproteins (P-gp) [3].

References:

[1] Moreau, P.; Karamanesht, II; Domnikova, N.; Kyselyova, M.Y.; Vilchevska, K.V.; Doronin, V.A.; Schmidt, A.; Hulin, C.; Leleu, X.; Esseltine, D.L.; et al. Pharmacokinetic, pharmacodynamic and covariate analysis of subcutaneous versus intravenous administration of bortezomib in patients with relapsed multiple myeloma. Clin Pharmacokinet 2012, 51, 823-829, doi:10.1007/s40262-012-0010-0.

[2] Reece, D.E.; Sullivan, D.; Lonial, S.; Mohrbacher, A.F.; Chatta, G.; Shustik, C.; Burris, H., 3rd; Venkatakrishnan, K.; Neuwirth, R.; Riordan, W.J.; et al. Pharmacokinetic and pharmacodynamic study of two doses of bortezomib in patients with relapsed multiple myeloma. Cancer Chemother Pharmacol 2011, 67, 57-67, doi:10.1007/s00280-010-1283-3.

[3] Hong, E.P.; Kim, J.Y.; Kim, S.H.; Hwang, K.M.; Park, C.W.; Lee, H.J.; Kim, D.W.; Weon, K.Y.; Jeong, S.Y.; Park, E.S. Formulation and Evaluation of a Self-microemulsifying Drug Delivery System Containing Bortezomib. Chem Pharm Bull (Tokyo) 2016, 64, 1108-1117, doi:10.1248/cpb.c16-00035.

4. Similarly, in section 2.5, the reasons on drug resistance of BTZ should be supplemented.

Response: Much appreciated for your suggestion on the cause of drug resistance of BTZ, which we have supplemented in the article: “Studies have shown that drug use results in mutations at key sites of PSMB5, which encodes the β5 subunit. This eventually leads to an increase in the β5 subunit, which cancels out the bound β5.”

5. In section 3.6, I suggest that the typical biomimetic nanomaterials should be introduced.

Response: At present, biomimetic nanomaterials such as protein polypeptides, DNA, RNA, exosomes, phages, various cells and cell membranes have been extensively studied. Among that, albumin, exosomes and cell membranes are most widely explored. Specifically, a variety of cell membranes have proven to be useful for drug encapsulation, such as erythrocyte membranes, platelet membranes, tumor membranes, macrophage membranes, neutrophils membranes, etc.

The description was updated in the revision and highlighted in red.

6. Some errors exist in the manuscript, and the full text needs to be checked thoroughly. For example:

1) In Table 2, the regimen “BOR” should be “BDR”.

2)The serial number of section “2.2. Properties and applications of BTZ” should be section “2”.

3) The name of the “NF-κB” should be written in consistent but not “NF-KB”

Response: Thanks for your correction and all above errors were revised. Meanwhile, we reviewed thoroughly the full text and corrected some similar errors.

Special thanks for your insightful comments and suggestions.